# Caveolae-Associated Protein 3 (Cavin-3) Influences Adipogenesis via TACE-Mediated Pref-1 Shedding

**DOI:** 10.3390/ijms21145000

**Published:** 2020-07-15

**Authors:** Phil June Park, Sung Tae Kim

**Affiliations:** 1Bioscience Laboratory, AMOREPACIFIC R&D Center, 1920 Yonggu-daero, Giheung-gu, Yongin-si, Gyeonggi-do 17074, Korea; 2Department of Pharmaceutical Engineering, Inje University, 197 Inje-ro, Gimhae-si, Gyeongsangnam-do 50834, Korea

**Keywords:** adipogenesis, 3T3-L1 adipocytes, Cavin-3, Pref-1

## Abstract

Abnormal adipogenesis regulation is accompanied by a variety of metabolic dysfunctions and disorders. Caveolae play an important role in the regulation of fat production, modulated by caveolae-associated proteins (Cavin-1 to 4). Here, we investigated the role of Cavin-3 in lipogenesis and adipocyte differentiation, as the regulatory functions and roles of Cavin-3 in adipocytes are unknown. A Cavin-3 knockdown/overexpression stable cell line was established, and adipogenesis-related gene and protein expression changes were investigated by real-time quantitative PCR and Western blot analysis, respectively. Additionally, confocal immune-fluorescence microscopy was used to verify the intracellular position of the relevant factors. The results showed that Cavin-3 mRNA and protein expression were elevated, along with physiological factors such as lipid droplet formation, during adipogenesis. Cavin-3 silencing resulted in retarded adipocyte differentiation, and its overexpression accelerated this process. Furthermore, Cavin-3 knockdown resulted in decreased expression of adipogenesis-related genes, such as *PPAR-γ, FAS, aP2*, and *Adipoq*, whereas preadipocyte factor-1 (*Pref-1*) was markedly increased during adipocyte maturation. Overall, Cavin-3 influences caveolar stability and modulates the tumor necrosis factor-alpha-converting enzyme (TACE)-mediated Pref-1 shedding process in both mouse and human adipocytes. The Cavin-3-dependent shedding mechanism appears to be an important process in adipocyte maturation, providing a potential therapeutic target for obesity-related disorders.

## 1. Introduction

Adipose tissues, which store energy in the form of lipids, and mobilize this energy in response to increased energy demands imposed by exercise, fasting, and hormonal stimulation, are important for maintaining energy balance in the body [1,2]. However, chronic excess energy intake relative to expenditure, and the consequent abnormal adipocyte differentiation and development of hyperplastic adipose tissue, can cause obesity and major health problems such as hypertension, heart disease, and diabetes mellitus [3,4,5,6,7]. Therefore, the regulation of adipocyte differentiation is crucial for the avoidance of obesity and related health problems.

Preadipocyte factor 1 (Pref-1) is highly expressed in preadipocytes and inhibits adipocyte differentiation by maintaining the preadipocyte state. Pref-1 is a transmembrane protein consisting of six epidermal growth factor (EGF)-like repeats, a juxtamembrane region, a single transmembrane domain, and a short cytoplasmic tail [8,9,10,11]. Full-length Pref-1 is proteolytically cleaved at its extracellular domain by tumor necrosis factor-alpha-converting enzyme (TACE, ADAM17), generating soluble forms of Pref-1 [12,13,14,15]. The soluble forms of Pref-1 produced by TACE are reported to inhibit adipocyte differentiation by suppressing factors involved in the early stage of adipogenesis, namely the CCAAT/enhancer-binding protein beta and delta (C/EBPβ and C/EBPδ), via induction of Sox9 expression through activation of the MEK/ERK pathway [15]. Pref-1 expression decreases gradually during adipocyte differentiation, and decreased Pref-1 expression alone is sufficient to increase adipocyte differentiation [11].

Caveolae are distinct features of fat cells, located in the cytosol, and comprised of small invaginations of lipid rafts from the plasma membrane, and play critical roles in various cellular processes such as the protection of adipocytes from nonionic detergent-dependent solubilization and/or toxic effects [16,17,18]. In metabolic regulation, for example, caveolae play important roles in insulin signaling and glucose transporter 4 trafficking in adipocytes [19]. Such caveolae consist of caveolins (Cav1, 2, and 3) and caveolae-associated protein groups (Cavin-1, 2, 3, and 4); each family is differentially associated with the biogenesis and function of caveolae. Some members of these protein families, such as Cav1, Cav2, and Cavin-1, are associated with adipocyte differentiation and biological properties [20,21,22,23].

Cavin-3, also known as serum deprivation response factor-related gene product that binds to c-kinase (SRBC) or protein kinase C delta binding protein (PRKCDBP), is an established serum deprivation response (SDR) gene [24]. Its gene product is highly localized in caveolae and contains a leucine zipper-like motif at its N-terminal region, essential for localization to caveolae, as well as two PEST sequences in its C-terminal region [25,26]. Cavin-3 mRNA is induced by serum starvation and has been detected in a wide variety of tissues and cultured cell lines. Furthermore, Cavin-3 has been reported to be inactivated in many cancers [27,28,29]. Despite its involvement in increase with adipocyte differentiation [30], the exact mechanism of action of Cavin-3 in adipogenesis has not been elucidated. Reports on the effect of Cavin-3 on the body fat content of knockout mice are conflicting [31,32], and the underlying molecular mechanisms by which Cavin-3 affects adipocyte differentiation remain unknown.

In this study, we found that Cavin-3 gene expression is elevated during 3T3-L1 adipocyte differentiation. To validate and investigate the mechanism of this change, we measured adipogenesis–related gene expression levels in Cavin-3 knockdown (siCavin-3) and overexpression (over-Cavin-3) cell lines. In particular, we focused on preadipocyte factor 1 (Pref-1) and its significant changes in siCavin-3 cell lines and developed a model for Pref-1-associated regulation of adipocyte differentiation.

## 2. Results

### 2.1. Physiological Effect of Cavin-3 in Knockdown or Overexpression Cell Lines

To investigate the association between Cavin-3 and adipocyte differentiation, Western blot analysis was performed to assess the expression of Cavin-3 during maturation of 3T3-L1 adipocytes (Figure 1A) [30]. After the induction of adipocyte differentiation, Cavin-3 levels gradually increased, and continued to do so until day 8.

To verify the functional role of Cavin-3 in adipocyte differentiation, we established Cavin-3 knockdown (siCavin-3) and overexpression (over-Cavin-3) cell lines and assessed adipocyte maturation in both systems. On day 8, lipid droplets (LDs), a physiological maturation marker in adipocytes, were visualized by Oil Red O (ORO) staining. Intracellular total triacylglycerol (TG) content was quantified by measuring the absorbance using Sudan II (490 nm). In siCavin-3, the number of stained LDs and intracellular TG content was significantly reduced compared to the control cells (Figure 1B). As shown in Figure 1C, over-Cavin-3 cells showed greater ORO staining than did mock-transfected cells on day 5 after induction of differentiation, but no significant difference was observed after full differentiation on day 8 (data not shown). The over-Cavin-3 cells had accumulated 27% more TG content than mock-transfected cells by day 5 (Figure 1C, lower panel). There was no significant difference in the total cell number of each group on day 8 after maturation (Figure 1D), demonstrating that each adipogenesis level was normalized under the same condition.

### 2.2. Effect of Cavin-3 Knockdown or Overexpression on Adipogenesis-Related mRNA Expression

Prior to the validation of adipogenesis-related mRNA expression, we confirmed Cavin-3 protein expression using Western blot analysis, to verify whether the Cavin-3 gene was clearly inhibited or promoted by gene manipulation. Protein expression of Cavin-3 clearly increased on day 8, compared to day 0 of differentiation in the control groups (normal 3T3-L1, Mock, and siControl). In over-Cavin-3 cells, Cavin-3 protein expression was significantly greater than in Mock cells on day 0, which increased by day 8. However, Cavin-3 protein expression was not detected in siCavin-3 cells at any time-point (Figure 2A).

The expression levels of Cavin-3 and adipogenesis-related genes, including preadipocyte factor 1 (*Pref-1*), peroxisome proliferator-activated receptor gamma (*PPARγ*) and adiponectin (*Adipoq*), fatty acid synthase (*FAS*), adipocyte protein 2 (*aP2*), and stearoyl-CoA desaturase (*SCD-1*) were determined by real-time quantitative polymerase chain reaction (RT-qPCR) during adipogenesis, respectively (Figure 2B–E). 

In siCavin-3 cells, Cavin-3 mRNA showed the initial inhibition was less than 30%, but markedly decreased approximately 70% after 8 days of differentiation (Figure 2B, left panel). Additionally, to determine whether the change of Cavin-3 affects adipogenesis-related gene expression, we verified the mRNA expression levels of *Pref-1, PPARγ, Adipoq, FAS*, and *aP2* at various time points after differentiation. The mRNA expression of *Pref-1* was gradually decreased during adipocyte differentiation in siControl cells, but sharply increased in siCavin-3 cells, even after the induction of differentiation (Figure 2B, center panel). In contrast, the mRNA expression levels of *PPARγ* and Adipoq, two well-known adipocyte differentiation markers [33,34], FAS and aP2 in the late stage of adipogenesis were suppressed in siCavin-3 cells (Figure 2B, right panel and C).

In over-Cavin-3 cells, Cavin-3 expression levels were significantly different, approximately 2.2 times (^††^
*p* < 0.01), between over-Cavin-3 and mock-transfected cells at the beginning of differentiation (day 0) (Figure 2D, left panel). However, the expression gap rates were gradually reduced and showed no difference by day 8. In contrast to siCavin-3 cells, the expression level of Pref-1 in over-Cavin-3 cells was similar to mock transfected cells (Figure 2D, center panel). Interestingly, *PPARγ, Adipoq, FAS* and *SCD-1* in over-Cavin-3 cells showed a rapid increase in gene expression during the differentiation period, unlike in the Mock cells (Figure 2D, right panel and E). While *PPARγ, Adipoq, FAS*, and *SCD-1* gene expression showed similar patterns in over-Cavin-3 and mock-transfected cells on day 2, they significantly increased (^†^
*p* < 0.05, ^††^
*p* < 0.01, ^†††^
*p* < 0.001) in over-Cavin-3 cells compared to mock cells thereafter. There was no statistical difference in ORO stained LDs and TG content (Appendix A) among the control cells, which included siControl, Mock, and non-transfected 3T3-L1 cells. Taken together, based on these results, it was confirmed that the stable cell lines promoting/suppressing Cavin-3 gene expression significantly regulated the increase/decrease of Cavin-3 gene expression, respectively. 

We also monitored the expression levels of CCAAT/enhancer-binding protein beta and delta (*C/EBPβ* and *C/EBPδ*) (Appendix A) [35,36,37]. Based on these additional results, it was confirmed that genes in later phases of adipocyte differentiation were also affected by Cavin-3 gene modulation. Overall, these results suggest that changes in Cavin-3 gene expression are linked to the regulation of adipogenesis-related gene expression.

### 2.3. Changes in Cavin-3 Gene Expression Influence the Cleavage of Pref-1

The aforementioned results showed that Cavin-3 modulation affects the mRNA expression of Pref-1, an early regulator of adipocyte differentiation [7,8,9,10]. A small fragment of Pref-1, produced by TACE through proteolytic cleavage (shedding), is a well-known adipogenesis inhibitor [11,12,13,14]. While previous studies have reported an association between Cavin-3 and TACE activity [26], the role of Cavin-3 in TACE-mediated Pref-1 shedding in adipocytes remained unclear.

The changes in TACE gene and protein expression during adipocyte maturation were confirmed, and were found unrelated to changes in Cavin-3 expression (Appendix A). To determine whether Cavin-3 is involved in controlling the TACE-mediated Pref-1 cleavage process, Pref-1 shedding results were validated using Western blot analysis in normal 3T3-L1 adipocyte and Cavin-3 modulated stable cell lines (Figure 3A). In particular, we focused on the expression level change of Pref-1 small fragment protein, which has known to be involved in the adipogenesis. As a result, the smaller band of Pref-1(21 kDa) was detected on day 0 in 3T3-L1 adipocytes, but, weakly present in 3T3-L1 adipocytes, siControl, Mock, and over-Cavin-3 at day 8. Interestingly, the small fragment of Pref-1 was significantly expressed and sustained in siCavin-3 at day 8 of maturation (Figure 3A, center panel). These results suggest that Cavin-3 may affect Pref-1 small fragment levels via TACE-mediated shedding during adipocyte maturation.

To further investigate the involvement of Cavin-3 in TACE-mediated Pref-1 cleavage, normal 3T3-L1 and Cavin-3 modulated cell lines were treated with the tissue inhibitor of metalloproteinase 3 (Timp3) peptide, a well-known TACE activity blocking agent [38,39], and assessed by ORO staining (Figure 3B and Appendix A) and TG content measurements (Figure 3C and Appendix A), respectively. The formation of LDs and the TG content increased in all cell types after 5 days of differentiation with timp3 treatment, but no significant changes were detected in over-Cavin-3 cells (Figure 3B, lower panel). To determine whether Cavin-3, as a physical access modulator, was involved in TACE-dependent Pref-1 shedding, all cell types were treated with timp3 and the levels of Pref-1 small fragments were assessed. In siCavin-3 cells, the small fragment of Pref-1 was undetectable by day 5 of differentiation (Appendix A, center panel). These results suggest that the association between Cavin-3 and TACE-mediated Pref-1 cleavage could be one feasible mechanism for controlling adipocyte differentiation

It has been previously reported that changes in Cavin-3 expression could undermine the structural stability of Cavin-1/Cav1 [40], and that Cavin-3 is also closely associated with extracellular signaling-regulated kinase (ERK) and protein kinase B (known as AKT) pathways [31]. To verify these findings, changes in ERK and AKT levels were measured in all cell lines at 8 days after differentiation (Figure 3D and Appendix A). While, the change in Cavin-3 expression had no significant effect on Cavin-1/Cav1 protein expression, ERK and AKT phosphorylation was reduced/increased according to the suppression/promotion of Cavin-3 expression, respectively. This is consistent with previous studies that show ERK phosphorylation promotes the differentiation of adipocytes [41].

### 2.4. Localization of Cavin-3 Might Affect TACE-Mediated Pref-1 Shedding

To validate how Cavin-3 affects TACE-mediated Pref-1 shedding, the localization and physiological changes of Cavin-3 and Pref-1 were observed using confocal immunofluorescence microscopy (Figure 4). As a result, Cavin-3 appeared to be sufficiently expressed in the differentiated adipocytes and located as if covering the stained lipid droplets in the cells. (Figure 4A). There was a noticeable difference in expression and localization of Pref-1 and Cavin-3 pre- and post-maturation (Figure 4B). On day 0, Pref-1 seemed to spread throughout the cell, except in the nucleus, and Cavin-3 was rarely expressed (Figure 4B, upper panel). On day 8, although no further positioning was confirmed using plasma membrane markers, Pref-1 seemed detected and appeared to be distributed outside the cell, as shown in the previous study about the location of full frame sized Pref-1 [12,13,14,15], even though the cell size was increased after differentiation (Figure 4B, lower panel). Notably, as Cavin-3 localization increased and concentrated around the lipid droplets, Pref-1 expression concentrated around the cell membrane.

To ensure that the distribution patterns of Pref-1 and Cavin-3 were equally applicable to human-specific adipocytes, changes before and after adipocyte differentiation were validated in human adipose derived stem cells (ADSCs) [42]. In ADSCs, after the induction of adipogenesis, Cavin-3 protein levels gradually increased over 21 days of differentiation (Appendix A) and the expression and distribution patterns of Pref-1 were similar to those observed in 3T3-L1 cells (Figure 4C, upper panel). Based on the results at day 21 of ADSC differentiation, there was no significant difference in Cavin-3 expression between ADSCs and 3T3-L1 cells (Figure 4C, lower panel). Therefore, our findings suggested that Cavin-3 is commonly associated with adipocyte maturation in both mouse and human adipocytes, and is likely an important component in this process. 

### 2.5. Cavin-3 Supports Caveolar Stability to Inhibit the Translocation of Pref-1 Small Fragment into Cytosol

We showed that Cavin-3 gene modulation is involved in TACE-mediated Pref-1 shedding, and also in spatial localization. According to previous studies, Cavin-3 regulates the functional and structural integrity of caveolae as a caveolin adapter protein [26], and caveolar stability has been reported to be a critical regulator of TACE activity [23]. Additionally, cholesterol-lowering drugs such as lovastatin have been reported to disrupt caveolar stability [43,44].

To determine whether the modulation of caveolar stability using disruptors affects adipogenesis-related factors, the caveloae group proteins were examined in normal 3T3-L1 adipocytes after treatment with 5 µM lovastatin (Figure 5A). The drug treatment results showed that although the increment of Cavin-3 protein expression depends on the adipogenesis process, the expression of caveolae stability-related and adipogenesis facilitated signals, such as Cavin-1, Cav-1, and PPARγ, decreased with increasing expression of Pref-1, at which time the TACE did not change. Although the results did not directly confirm the disrupting caveolae formation by lovastatin treatment, previous study has shown that changes in the caveolar stability could occur due to the disruption of caveolae formations caused by a drug like lovastatin [43]. Therefore, we assume that changes in the caveolar stability by lovastatin could induce changes in adipogenesis-related factors such as Pref-1 and PPARγ.

Next, to clarify whether the change of caveolar stability affected Cavin-3, confocal immunofluorescence microscopy was used to investigate changes in localization in lovastatin-treated over-Cavin-3 cells. The disruption in caveolae stability resulted in Cavin-3 expression being dispersed throughout the cytoplasm. Furthermore, Cavin-3 and Pref-1 did not co-localize (Figure 4B, upper panel, Figure 5B and Appendix A). These results suggest that Cavin-3 modulation by gene manipulation affects caveolar stability by influencing the support role of Cavin-3, thus affecting the regulation of Pref-1 small fragment translocation into the cytosol (Figure 6).

## 3. Discussion

Adipose tissues serve as major organization centers for the maintenance of homeostasis in the human body. Adipose tissue and inflammation are closely related, and imbalances can lead to metabolic dysfunction and disorders when differentiation and regulation are not controlled. Therefore, the proper regulation of fat cell differentiation represents an important strategy for maintaining a healthy state. It is of high importance to conduct an in-depth study into the correlation between fat production-related factors during adipocyte maturation.

Previous studies have shown that dramatic changes in body fat in vivo occur as a result of the silencing of Cavin-3 [31,32]; however, the precise role of Cavin-3 had not been elucidated [16,17,18,19,20,21,22,23]. The results of the present study show that the promotion or inhibition of Cavin-3 affects adipocyte differentiation (Figure 1B). To better understand the underlying mechanisms, we verified our results through cellular and molecular approaches, which included the investigation of adipogenesis-related gene expression (*Pref-1, PPARγ, Adipoq, FAS, aP2*, and *SCD-1*; Figure 2B,C and Appendix A).

It has been reported that Pref-1 is highly expressed in an undifferentiated state and inhibits adipogenesis by maintaining a preadipocyte state [7,8,9,10]. Full-length Pref-1 is proteolytically cleaved at its extracellular domain by TACE, generating a soluble form of 21 kDa Pref-1 [11,12,13,14], which inhibits adipocyte differentiation by suppressing various factors. Such factors include C/EBPβ and C/EBPδ, both of which are involved in the early stages of differentiation (Appendix A). Although it decreases after the initial 15 min of differentiation and increases after a certain period of time, Pref-1 cleavage fragments also induce Sox9 expression through activation of the MEK/ERK pathway [45]. Pref-1 expression gradually declines during adipocyte differentiation, and the reduction in expression is sufficient to stimulate adipocyte differentiation [13,14]. Notably, Pref-1 dependent regulation via Cavin-3 has been suggested as a possible mechanism.

To prove the possibility of Cavin-3-dependent Pref-1 regulation, we investigated the correlation among TACE, Cavin-3, and Pref-1 in Cavin-3 modulated stable cell lines. Inhibition of TACE activity using timp3 caused an expected change in TACE-mediated Pref-1 shedding, which confirmed that TACE operates normally and is not the direct reason of abnormal differentiation in siCavin-3 and over-Cavin-3 cells (Figure 3B,C). These results convinced us that there was no specific abnormality in the activation of TACE in both siCavin-3 and over-Cavin-3. Previous studies reported that Cavin-3 dictates the balance between the ERK and AKT signaling in Cavin-3-knockout human fibroblast cell line, SV589, and mouse lung tissue [31]. However, our results showed that the expression of pERK and pAKT was simultaneously inhibited in siCavin-3 cells and increased in over-Cavin-3 cells (Figure 3D). In addition, the expression of pERK was increased in all Cavin-3 gene modified cells at the early stage of differentiation (15 min), with marked differences in expression levels (Appendix A). These findings were consistent with findings of a previous study [41]. Nevertheless, some limitations exist in this experimental outcome statement, including the fact that there may be subtle differences in cell activity among groups in a given single experiment, because the TACE activity itself could not be directly measured. Therefore, additional research is required for a complete understanding of the correlation between Cavin-3 and TACE on Pref-1 shedding. 

Several studies on the association between caveolae groups and adipocyte differentiation have shown that caveolae stability-dependent caveolin-1 expression plays an important role in the maintenance of TACE activity [43,44]. Based on this, lovastatin, known as a caveolae stability inhibitor, was used to determine whether stability was altered in normal 3T3-L1 and over-Cavin-3 manipulated cells. When lovastatin was used to treat normal 3T3-L1 cells, the expression of adipogenesis-related factors and caveolae group genes, such as Cavin-1/Cav1 was also simultaneously regulated along with the associated physiology (inhibition of adipogenesis), although the expression of TACE did not change as Cavin-3 gene expression increased during the maturation process (Figure 5A). Based on these data, we believe that the shedding process of Pref-1 might be due to existing TACE, although it caused abnormalities in the caveolae group stability by lovastatin. In addition, lovastatin has been already reported as an adipocyte differentiation inhibitor [42], and the inhibition mechanism of adipogenesis by lovastatin could suggest that the opportunity of TACE-mediated Pref-1 shedding is further provided to inhibit the differentiation process, because of the suppression of caveolae stability via gene expression inhibition of the caveolae group, such as Cavin-1 and Cav1. 

Cavin-3 is localized at the cell membrane during the process of adipocyte differentiation (Figure 3A). It has previously been reported that the proteolytic cleavage (shedding) of Pref-1 starts with full length Pref-1 located at the cell membrane [7,11], suggesting that the expression and localization changes of Cavin-3 appear to affect the location of the Pref-1 small fragment. Moreover, increased Cavin-3 could possibly directly inhibit TACE-mediated Pref-1 cleavage, as the small fragment of Pref-1 was undetectable in overexpressing Cavin-3 cells, such as mature adipocyte or over-Cavin-3 cells (Figure 3A). However, the clear part identified by the results is that Cavin-3 serves as the stability support for the caveolae group to regulate the role of TACE, along with the translocation inhibition of Pref-1 small fraction into the cytosol. In this aspect, it was necessary to confirm whether Cavin-3-dependent changes of caveolae stability caused unexpected differentiation abnormalities. Through the lovastatin treatment in over-Cavin-3 cells, it was found that Pref-1 and Cavin-3 had similar dispersal patterns to those of undifferentiated cells (Figure 5B). As a result, lovastatin treatment during adipogenesis regulated the stability of the caveolae group to lower the expression of Cavin-1/Cav1, thereby recruiting Pref-1 shedding by TACE to inhibit the adipogenesis process, irrespective of the increased Cavin-3 expression.

Taken together, our results suggest the following four steps: (1) Cavin-3 expression increases throughout adipogenesis; (2) the stability of the caveolae groups is supported by increased Cavin-3; (3) increased Cavin-3 inhibits translocation of Pref-1 small fragment into the cytosol; (4) adipocyte differentiation is promoted through the inhibition of Pref-1 expression (Figure 6). Based on these findings, Cavin-3 might be used as an early marker of adipocyte differentiation, or further, a regulatory target for obesity-related disorders.

## 4. Materials and Methods

### 4.1. Cell Culture and Differentiation

Mouse 3T3-L1 preadipocytes (ATCC, CL-173) were incubated in Dulbecco’s modified Eagle’s medium (DMEM; Lonza, Walkersville, MD, USA) containing 10% calf serum (Thermo Fisher Scientific, Waltham, MA, USA) in a humidified incubator with 5% CO_2_. For the differentiation experiments, medium was replaced on the third day with DMEM containing 10% fetal bovine serum (FBS; Thermo Fisher Scientific), 10 µg/mL insulin (Sigma-Aldrich, St. Louis, MO, USA), 0.5 mM 3-isobutyl-1-methylxanthine (IBMX; Sigma-Aldrich, St. Louis, MO, USA), and 1 µM dexamethasone (DEX; Sigma-Aldrich, St. Louis, MO, USA). After 2 days, the medium was changed to DMEM containing 10% FBS with 10 µg/mL insulin, and the cells were incubated for 3 additional days. The cells were then cultured in DMEM containing 10% FBS without insulin for 3 days.

Human adipose derived stem cells (ADSCs; Lonza, Walkersville, MD, USA) were cultured in ADSC growth medium in a humidified incubator with 5% CO_2_. To induce differentiation, the ADSCs were incubated in DMEM (Lonza, Walkersville, MD, USA) containing 10% FBS (PAA; Pasching, Austria), 10 µg/mL insulin (Sigma-Aldrich, St. Louis, MO, USA), 0.5 mM IBMX (Sigma-Aldrich, St. Louis, MO, USA), 1 µM DEX (Sigma-Aldrich, St. Louis, MO, USA), and 1 µM troglitazone (Sigma-Aldrich, St. Louis, MO, USA), with replenishment of the medium every 2 days.

### 4.2. Chemicals

TACE inhibitor, metalloproteinase inhibitor 3 (Timp3), and caveolae stability inhibitor, lovastatin, were purchased from Abcam (Eugene, OR, USA) and Sigma-Aldrich, respectively.

### 4.3. RNA Extraction and RT-qPCR

Total RNA was extracted using Reliaprep RNA cell miniprep system (Promega, Madison, WI, USA) according to the manufacturer’s instructions. Complementary DNA (cDNA) was synthesized from ≈1 µg of total RNA using Revertaid First cDNA Synthesis Kit (Thermo Fisher Scientific, Waltham, MA, USA). The cDNA was used as a template for RT-qPCR, performed using a 7500 Fast Real-Time PCR System (Life Technologies, Austin, TX, USA). Pre-designed TaqMan probe sets were purchased from Thermo Fisher Scientific (Waltham, MA, USA); their sequences are listed in Appendix A.

### 4.4. RNAi Experiment

To suppress the expression of the Cavin-3 gene, target regions in the open reading frame (ORF) of mouse Cavin-3 (mCavin-3; GenBank Accession No. NM_028444) were used to design a siRNA directed against Cavin-3. This siRNA was purchased as the duplex-ready stable form from Integrated DNA Technologies (IDT; Coralville, IA, USA). The sequences of the sense and antisense strands of siRNA directed against Cavin-3 were 5′-GGAGCUUUCAGCCUAAUUUtt-3′ and 5′-AAAUUAGGCUGAAAGCUCCtc-3′, respectively. The constructed siRNA was inserted into the BamHI/HindIII site of the pSilencer 2.1-U6 puro plasmid (Ambion, Inc., Austin, TX, USA). A scrambled fragment with no similarity to any mRNA listed in GenBank was used as a negative control (IDT). Preadipocyte states of 3T3-L1 were transfected with pSilencer 2.1-U6 puro-mCavin-3 RNA (siCavin-3) or with a pSilencer 2.1-U6 puro negative control (siControl) using FuGene 6 Transfection Reagent (Roche Diagnostics, IN, USA). The transfected cells were selected with puromycin (1.5 µg/mL, Sigma-Aldrich, St. Louis, MO, USA).

### 4.5. Overexpression Experiment

To produce stable Cavin-3-overexpressing cell lines, an mCavin-3 cDNA fragment was obtained by RT-PCR using 3T3-L1 poly A+ mRNA and the primers 5′-GGAATTCCAATGGGGGAGAGC-3′ and 5′-GGGTACCGGCTGCGCTCTCTAT-3′. The mCavin-3 cDNA was inserted between the XbaI and BamHI sites of the pcDNA3.1 vector (Invitrogen, CA, USA). 3T3-L1 cells were transfected with pcDNA3.1-U6 puro negative control (mock) and pcDNA3.1-U6 puro-mCavin-3 (over-Cavin-3) using FuGene 6 Transfection Reagent (Roche Diagnostics, Indianapolis, IN, USA). Cells stably expressing Cavin-3 were selected by puromycin (3 µg/mL) resistance.

### 4.6. Oil Red O Staining

Oil Red O (ORO; Sigma-Aldrich, St. Louis, MO, USA) stock solution was prepared by dissolving ORO in 60% propylene glycol (PG) (Santa Cruz Biotechnology, Santa Cruz, CA, USA), and used to highlight the amount of lipid droplet in differentiated 3T3-L1 adipocytes. On 0, 2, 5, and 8 days after differentiation induction, 3T3-L1 cells were washed twice with cold phosphate-buffered saline (PBS; Welgene, Daegu, Korea), before being fixed with 3.7% formaldehyde (Sigma-Aldrich, St. Louis, MO, USA) in PBS for 1 h. The fixed cells were washed with PG and stained with ORO working solution (0.3% Oil Red O in 60% PG) for 30 min. The cells were then washed three times with 85% PG and rinsed gently with tap water. The phenotypic changes in the cells were visualized and photographed using an Olympus IX71 microscope (Tokyo, Japan).

### 4.7. Analysis of TG Content

The amount of accumulated TG was measured using the Sudan II (Sigma-Aldrich, St. Louis, MO, USA) staining method. The cells were washed twice in cold PBS, before being fixed with 3.7% formaldehyde for 20 min. Cells were then stained with 0.5% Sudan II in 60% isopropyl alcohol (Sigma-Aldrich, St. Louis, MO, USA) for 1 h. The stained cells were washed gently with 70% ethyl alcohol (Sigma-Aldrich), and the TG-bound Sudan II was extracted with 4% NP-40 (Sigma-Aldrich, St. Louis, MO, USA) in isopropyl alcohol for 20 min. A Synergy H2 spectrophotometer (BioTek, Winooski, VT, USA) was used to measure the extract absorbance at 490 nm.

### 4.8. Western Blot Analysis

Proteins were resolved by sodium dodecyl sulfate-polyacrylamide gel electrophoresis (SDS-PAGE), transferred to membranes, and probed with the following antibodies: anti-Cavin-3 (Thermo Scientific Co., Waltham, MA, USA); anti-Pref-1, anti-Cavin1, and anti-Cav1 (Cell Signaling Technology, Danvers, MA, USA); anti-PKAα and anti-phospho-PKA α/β/γ (Santa Cruz Biotechnology, Dallas, TX, USA); anti-ERK1/2 and anti-phospho-ERK1/2 (R&D systems, Minneapolis, MN, USA); and anti-gamma tubulin (γ-TUB; Sigma-Aldrich, St. Louis, MO, USA). Quantitative analysis of all protein levels is presented in Appendix A.

### 4.9. Confocal Immunofluorescence Microscopy

3T3-L1 adipocytes were cultured on Lab-Tek chamber slides (Nunc, Rochester, NY, USA), fixed with 3.7% formaldehyde in PBS, permeabilized with 0.1% Triton X-100 in PBS containing 1% BSA for 5 min, and incubated with antibodies at 4 °C overnight. Alexa Fluor 488-conjugated donkey anti-goat IgG and Alexa Fluor 594-conjugated goat anti-rabbit IgG antibodies (Life Technologies, Austin, TX, USA) were used as secondary antibodies. The antibody-treated specimens were stained with Sudan III (Wako, Osaka, Japan) dissolved in 70% ethanol to visualize lipid droplets. DNA was stained with 4′,6-diamidino-2-phenylindole. The specimens were mounted, and images were obtained using a confocal laser scanning microscope (LSM 700; Carl Zeiss, Jena, Germany).

### 4.10. Statistical Analysis

All results are expressed as the mean ± standard deviation (SD). Differences between groups were calculated using the Student’s unpaired *t*-test. Differences with *p* < 0.05 were considered statistically significant.

## 5. Conclusions

Cavin-3 expression is significantly elevated during adipocyte differentiation. Increased expression levels of Cavin-3 can promote both adipocyte differentiation and adipogenesis-related factors, and a low level of Cavin-3 reduces these effects. Cavin-3 plays an important role in adipogenesis by downregulating TACE-mediated shedding of Pref-1, a well-known preadipocyte maintenance factor.

## Figures and Tables

**Figure 1 ijms-21-05000-f001:**
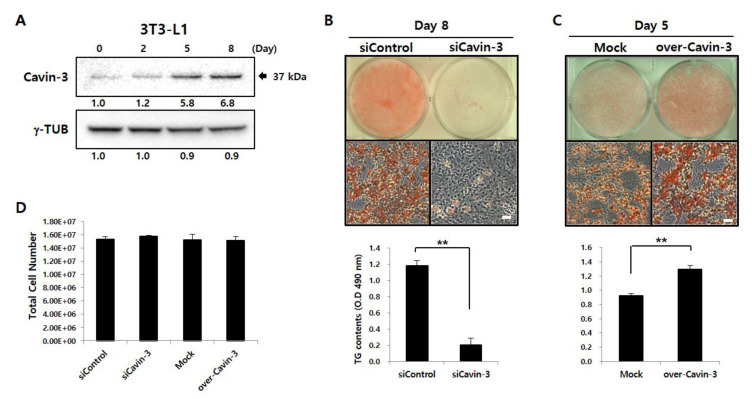
Caveolae-associated protein 3 (Cavin-3) profiling during adipogenesis and Cavin-3 based control of adipocyte differentiation. (**A**) Expression of Cavin-3 protein during adipocyte differentiation. 3T3-L1 cells were differentiated for 8 days and harvested at various time points (days 0, 2, 5, and 8). Anti-gamma tubulin (γ-TUB) antibody was used as a control and the grouping of blots cropped from different gels. (**B**, **C**) The Oil Red O (ORO) staining and total glyceride (TG) content were validated in siCavin-3 and over-Cavin-3 cells after 8 and 5 days of differentiation, respectively. Scale bar = 200 µm. The TG content was quantified by Sudan II staining the cells (*n* = 3) by measuring the absorbance at 490 nm. The data are presented as the means ± SD. (* *p* < 0.05, ** *p* < 0.01; data were analyzed with an unpaired Student’s *t*-test). (**D**) Total cell number was counted in siControl, siCavin-3, Mock, and over-Cavin-3 cells at day 8 of differentiation. The data are presented as the means ± SD.

**Figure 2 ijms-21-05000-f002:**
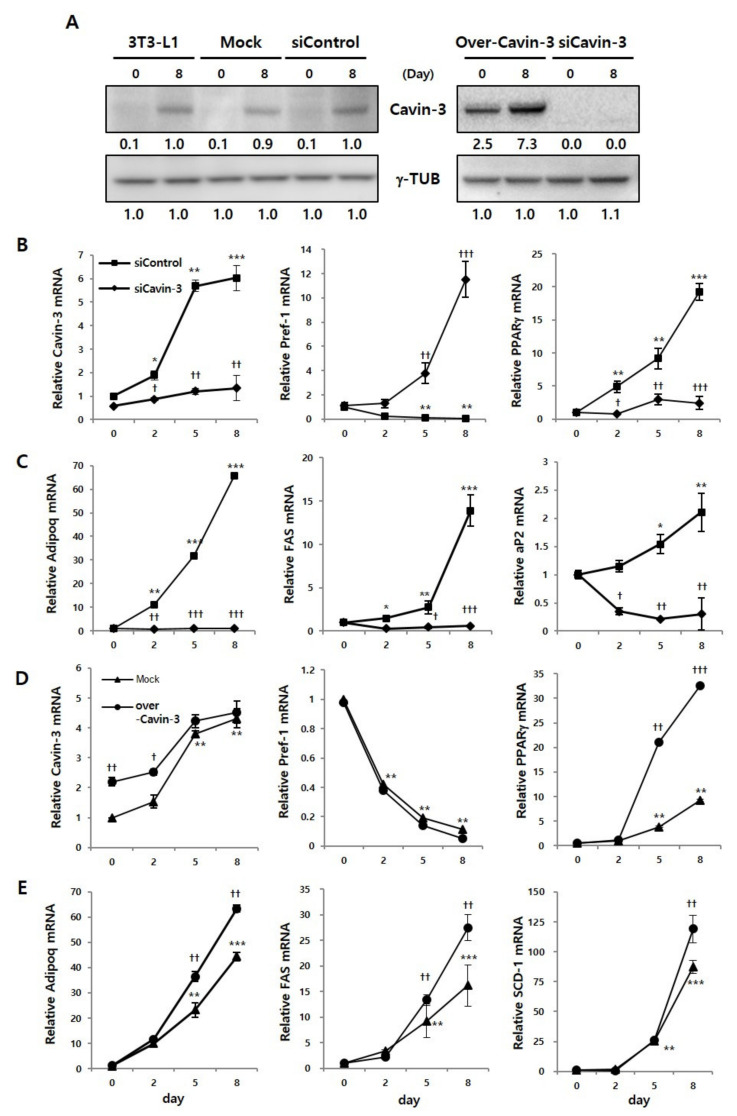
Cavin-3 gene modulation influences adipogenesis-related mRNA expressions. (**A**) Expression of Cavin-3 protein was validated in 3T3-L1 adipocytes, siControl, siCavin-3, Mock, and over-Cavin-3 cells by Western blot analysis at days 0 and 8 of differentiation. γ-TUB was used as a control and the grouping of blots cropped from different gels. (**B**–**E**) The mRNA expression of Cavin-3, preadipocyte factor 1 (*Pref-1*), peroxisome proliferator-activated receptor gamma (*PPARγ*), adiponectin (*Adipoq*), fatty acid synthase (*FAS*), adipocyte protein 2 (*aP2*), and stearoyl-CoA desaturase (*SCD-1*) in siControl (*n* = 3) vs. siCavin-3 (*n* = 3) and Mock (*n* = 3) vs. over-Cavin-3 (*n* = 3) cells, respectively. The expression levels were determined by RT-qPCR at various time points (0, 2, 5, and 8 days of differentiation). The data are presented as the means ± SD. (* *p* < 0.05, ** *p* < 0.01, *** *p* < 0.001 compared to siContol or Mock at day 0: ^†^
*p* < 0.05, ^††^
*p* < 0.01, ^†††^
*p* < 0.001 compared to siControl or Mock at each time point; data were analyzed with an unpaired Student’s *t*-test).

**Figure 3 ijms-21-05000-f003:**
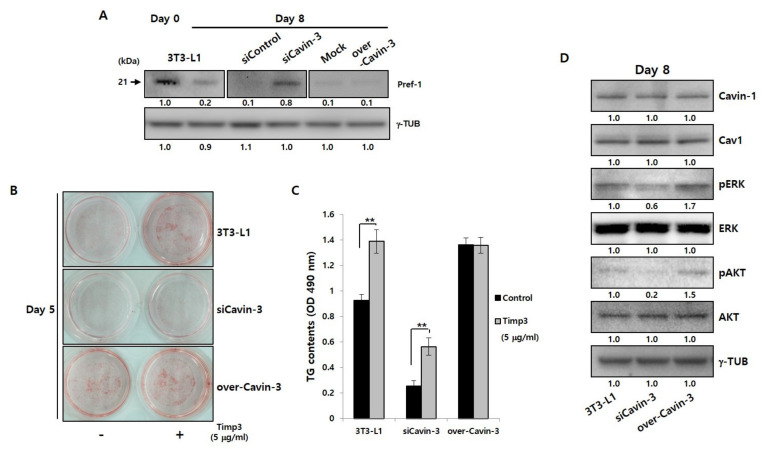
Cavin-3 gene modulation influences the Pref-1 shedding process, but not the caveolae group. (**A**) Western blot analysis of Pref-1 protein expression in normal 3T3-L1 adipocytes, siControl, siCavin-3, Mock, and over-Cavin-3 cells using specific Pref-1 antibody. Total protein was prepared before and 8 days after adipogenesis induction. γ-TUB was used as a control. (**B**) The effect of TACE activity blocking agent, timp3, affect adipocyte differentiation in normal 3T3-L1 adipocytes, siCavin-3, and over-Cavin-3 cells after 5 days of differentiation and analyzed by ORO staining. (**C**) The TG content of normal 3T3-L1, siCavin-3, and over-Cavin-3 cells after 5 days of differentiation with or without timp3 treatment. The intracellular TG content of cells in 6-well plates was measured (*n* = 3). The data are presented as the means ± SD. (* *p* < 0.05, ** *p* < 0.01; data were analyzed with unpaired Student’s *t*-test). (**D**) Caveolae group and related signaling protein levels were determined by Western blot analysis using specific antibodies. Total protein was prepared at day 8 after adipogenesis induction. γ-TUB was used as a control and the grouping of blots cropped from different gels.

**Figure 4 ijms-21-05000-f004:**
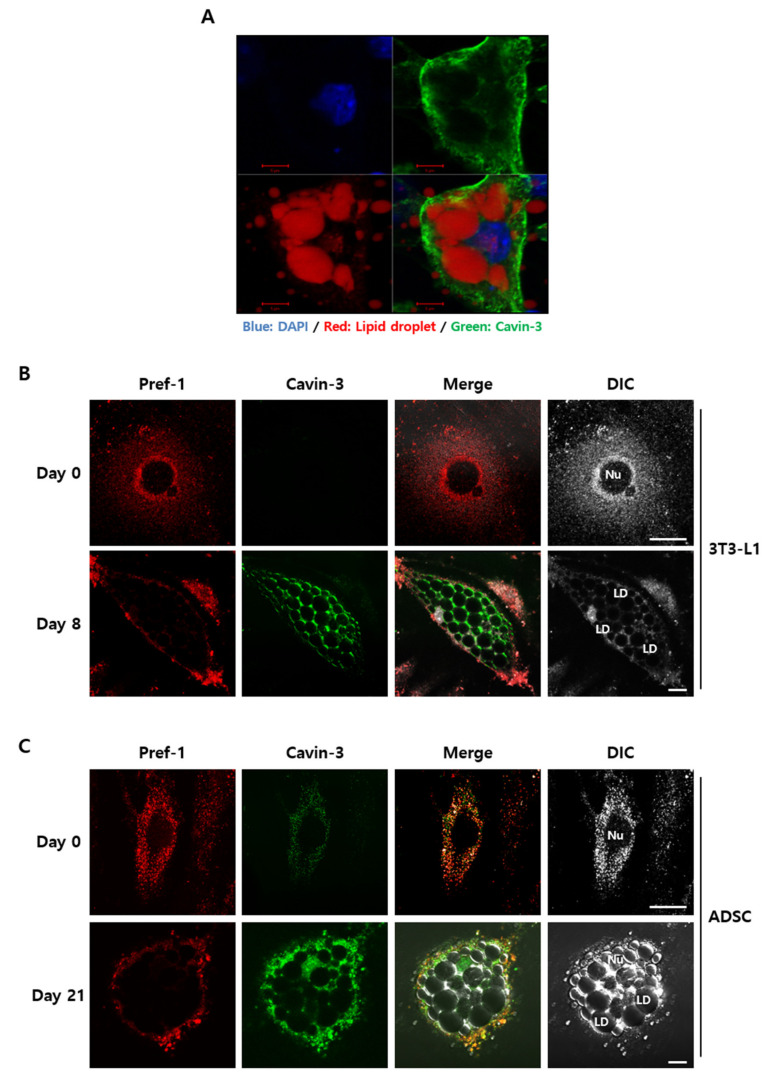
The exchange of Cavin-3 and Pref-1 subcellular localization during adipocytes differentiation in 3T3-L1 and human adipose derived stem cells (ADSC). The cells were fixed and stained with specific antibodies, respectively. (**A**) The nucleus was stained with 4,6-diamidino-2-phenylindole (DAPI) (blue), lipid droplets were stained with Sudan III (red), and Cavin-3 were stained with anti-Cavin-3 (green) antibody at day 8 of differentiation. (**B**,**C**) Normal 3T3-L1 adipocytes and human adipose-derived stem cells (ADSCs) were fixed with anti-Pref-1 (red) and anti-Cavin-3 (green) antibodies at each time point. Cells were imaged with a confocal microscope. Scale bar = 5 µm.

**Figure 5 ijms-21-05000-f005:**
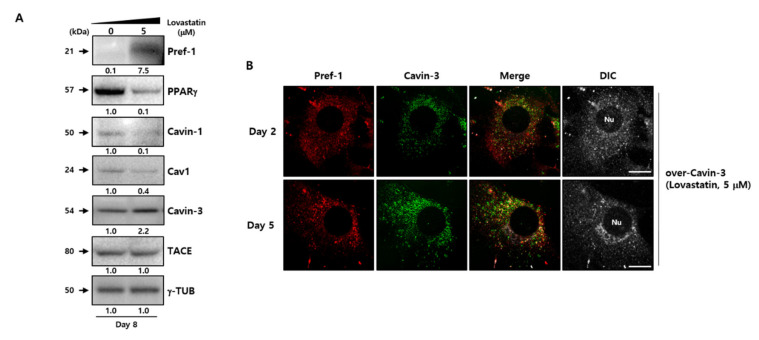
Cavin-3-dependent caveolar instability affects the TACE-mediated Pref-1 shedding process. (**A**) Specific protein levels were determined by Western blot analysis using specific antibodies during adipocyte differentiation with or without caveolar stability inhibitor, lovastatin, in normal 3T3-L1 adipocytes. γ-TUB was used as a control and the grouping of blots cropped from different gels. (**B**) Expression and subcellular localization of Pref-1 and Cavin-3 in over-Cavin-3 cells were validated during adipogenesis with or without lovastatin. The cells were fixed and stained with anti-Pref-1 (red) and anti-Cavin-3 (green) antibodies and observed with a confocal microscope. Scale bar = 5 µm.

**Figure 6 ijms-21-05000-f006:**
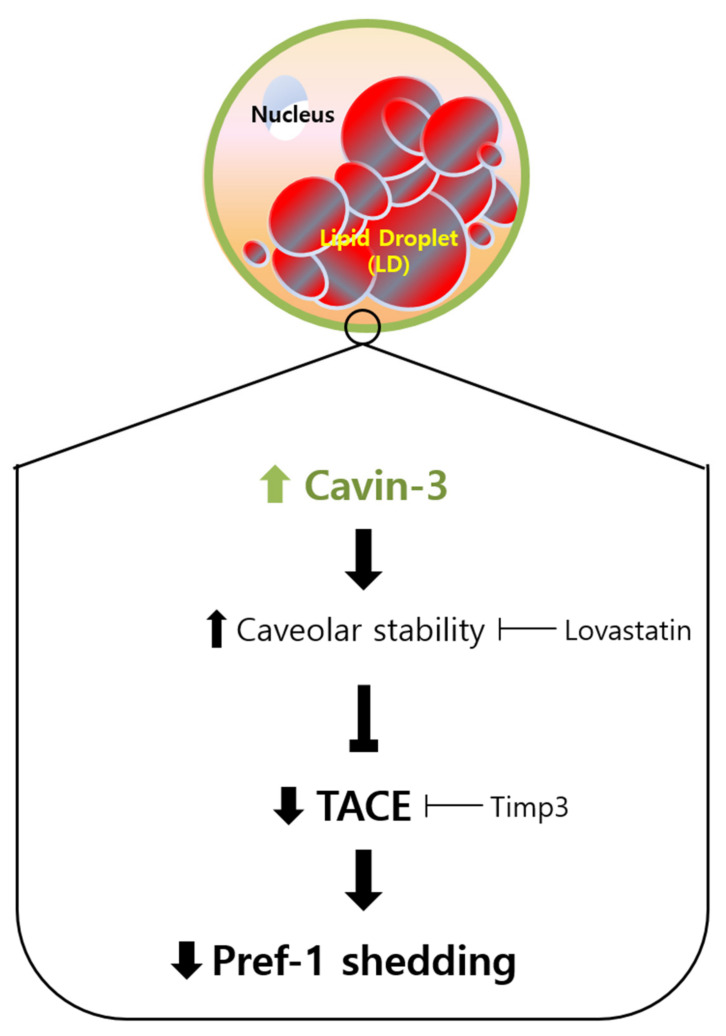
Schematic representation of the proposed mechanism for Cavin-3-mediated adipogenesis. Increased expression of Cavin-3 during the maturation of adipocytes induces caveolar stability, which in turn suppresses TACE activity involved in the Pref-1 shedding process, leading to the adipocyte differentiation.

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
