# Peer review of "Caveolae-Associated Protein 3 (Cavin-3) Influences Adipogenesis via TACE-Mediated Pref-1 Shedding"

_ijms, 2020, doi:10.3390/ijms21145000_

Round 1
Reviewer 1 Report
Using gene knockdown and over-expression in adipocyte cell lines, this study examines a role for TACE-mediated cleavage of Pref-1 as a downstream mediator of cavin-3's effects on pre-adipocyte differentiation. While most of the data presented are sound, there are several issues that the authors will need to address in a revised manuscript:-
Major issues (in no particular order)
- In several instances (Fig. 1A, 2A), it is problematic comparing effects of knockdown/overexpression using composite blots, due to potential differences in exposure times and other factors. It would therefore be helpful if data showing efficient knockdown/overexpression could be shown in a single gel/blot.
- Were all the experiments performed on single stable knockdown/overexpression clonal 3T3-L1 cell lines? Ideally, key findings should be replicated in at least one more clonal cell line of each, to ensure that the findings are not clonal artefacts.
- Lovastatin treatment - the authors need to show by TEM that this treatment actually reduces the number of caveolae in their hands.
- I appreciate it is not the thrust of the manuscript, but it would be helpful if the blots for cavin-1, Pref-1, P-ERK and P-Akt could be quantified with the appropriate statistics.
Minor points (in no particular order)
- Figs 3B, S2C - the images of ORO staining are very faint and do not look obviously representative of the quantitative data, so these need to be modified to improve clarity.
- Fig 4B - are the authors arguing for Pref-1/cavin-3 colocalisation? I ask as I am colourblind and so any relevant co-staining is not visible to me, and the text is not clear. If they are arguing for co-localisation, they should more explicitly state it in the text and include a Pearson correlation coefficient analysis.
Author Response
Response to Reviewer’s comment 1
Using gene knockdown and over-expression in adipocyte cell lines, this study examines a role for TACE-mediated cleavage of Pref-1 as a downstream mediator of cavin-3's effects on pre-adipocyte differentiation. While most of the data presented are sound, there are several issues that the authors will need to address in a revised manuscript:
Major issues (in no particular order)
Q1. In several instances (Fig. 1A, 2A), it is problematic comparing effects of knockdown/overexpression using composite blots, due to potential differences in exposure times and other factors. It would therefore be helpful if data showing efficient knockdown/overexpression could be shown in a single gel/blot.
A1. Thank you for your comment. We also agree with your opinion. As requested, we revised and added data showing efficient knockdown/overexpression in a single gel/blot, which were highlighted in our manuscript. However, we didn’t revise figure 1A because all blots were already shown in a single gel. So, we revised western blots in Figure 2A and displayed blots in a single gel as follows;
[Original version] [Revised version]
Q2. Were all the experiments performed on single stable knockdown/overexpression clonal 3T3-L1 cell lines? Ideally, key findings should be replicated in at least one more clonal cell line of each, to ensure that the findings are not clonal artefacts.
A2) Thank you for your comment. We also agree with your opinion. Prior to the experiment, we established three stable cell lines each. Among them, we monitored the cavin-3 associated genes and markers such as Pref-1 and PPARg. All cell lines showed the similar patterns in their expressions more or less; we attached law data related to expression levels as attached below. Based on law data, we chose the cell line (#1) and performed additional experiment.
[Additional Figure 1]
Q3. Lovastatin treatment - the authors need to show by TEM that this treatment actually reduces the number of caveolae in their hands.
A3. Thank you for your comment. As mentioned, we also considered an appropriate way to show the reduction of caveolae number; we attempted biochemical approaches. However, we couldn’t obtain a clear result supporting a change of caveolae level after treatment of lovastatin. According to the previous literature (Jansen et al., reference #43), they reported that a cholesterol substitution including lovastatin treatment induces the changes of composition and morphology of caveolae vesicles, and it was mentioned that the mobility and endocytosis function of caveolae could be changed. Hence, the article showed that caveloae number was reduced by lovastatin treatment as shown in Figure 5A, although they did not comment on manuscript.
As requested, we also agree with reviewer’s opinion; a transmission electron microscope (TEM) approach can provide a more clarified data. So, we attempted to prepare a sample for cell TEM measurement, however we stopped. It was not practically available to use cell TEM due to the COVID-19 issue. Since the TEM is not in Amorepacific laboratory as well as Inje University, we need to use it in different institutions. However, we can’t access and use it all in this situation; we really feel sorry for this situation. Based on our data and previous literatures, we believe that lovastatin leads to the suppression of differentiation through the level as well as composition of caveolae regardless of Cavin-3 increase. Our findings suggest a possibility that Cavin-3 changes are likely to affect fat differentiation through TACE-Pref-1 regulation through the effect of caveolae. We apologize for not providing the TEM data for a better understanding.
Q4. I appreciate it is not the thrust of the manuscript, but it would be helpful if the blots for cavin-1, Pref-1, P-ERK and P-Akt could be quantified with the appropriate statistics.
A4. Thank you for your comments. As mentioned, we also clarified the expression levels. To do so, we first prepared samples at day 8 after differentiation and performed western blot analysis of each sample. As results, we obtained each blot for cavin-1, Pref-1, P-ERK and P-Akt, which was quantitatively analyzed by Image J program developed at the National Institutes of Health and the Laboratory for Optical and Computational Instrumentation. Depending on band intensity, all data were calculated and statistically compared as follows; for a relative comparison, we set a Cavin-1 to the reference intensity 1 and compared to expression levels of other proteins. These results were added in Supplementary Information (Figure S4D).
[Additional Figure 2]

Reviewer 2 Report
In the present study, Park PJ and colleagues analyzed the role of Cavin-3, a caveolae-associated protein, on lipogenesis and adipogenesis. Furthermore, Cavin-3 modulation of TACE-mediated Pref-1 shedding was explored by the authors. For this, authors developed 2 cell lines of 3T3-L1 adipocytes, one with silencing of Cavin-3 expression and the other overexpressing the protein. Overall, this is an important and interesting subject, but the study is not carefully presented, and some conclusions taken by the authors are not fully supported by the experimental data.
Comments:
- In Introduction section essential information about Pref-1 and TACE is lacking.
- Page 2, lane 29: authors state that Cavin-3 levels reach a maximum at day 5 of differentiation and levels are sustained until day 8. However, Fig1A and the western-blotting quantification presented by the authors show more cavin-3 at day 8 comparing to day 5.
- The characterization of the siCavin-3 and over-Cavin-3 cell lines is described on page 3, lanes 13-28 and Fig2A. This should be presented behind data on fig 1B-D, that already show results with those cell lines. Additionally, no results of Fig 2A are described on the text, however RT-qPCR included in supplementary data are fully described. I suggest including Supp Fig S1 A and B on the manuscript.
- Fig2A: normalization with housekeeping gene is lacking
- Regarding the cell line overexpressing cavin-3, RT-qPCR data analysis of Fig Supp S1B show no differences on mRNA expression at day 5 and 8. Opposite results were obtained by WB. Authors should comment, please.
- The statistical analysis of all data should be clarified. It is not defined if the comparisons are performed between the day 0 and the different time points of the same condition or between different conditions at the same time point.
- Pref-1 is synthesized as a transmembrane protein but processed to generate soluble forms, including a large 50-kDa soluble form and additional small soluble forms. Several papers claim that only the large soluble form, but not the small soluble or the transmembrane forms of Pref-1, is biologically active to inhibit adipogenesis. Do authors have evidence of biological role of the 21kDa form of Pref-1 detected on the manuscript? Did the authors also detect the 50kDa form?
- Timp3 action should be validated by analysis of Pref-1 cleavage.
- Page 6, lane 18: “Cavin-3 is located at the inner membrane of mature adipocytes”. What inner membrane are the authors referring to? Furthermore, the immunofluorescence images of cavin-3 (Fig4) suggest a location surrounding the lipid droplets or in the cytosol of the adipocyte. The location at the cell membrane is not clear. The authors could perform a co-localization study using a classic cell membrane marker.
- By WB, Pref-1 is almost undetectable at day 8 of differentiation. In fact, it is known that Pref1 expression inhibition is crucial for the adipocyte differentiation. However, in Fig4 Pref-1 immunofluorescence detection levels are similar at day 0 and day 8. The authors should comment, please. In addition, Pref-1 location does not seem to be the plasma membrane. Again, a co-localization study with a plasma membrane marker could help.
- Authors state that caveolar stability is dependent on cavin-3. This manuscript does not include data supporting this result.
- In general, the sentences are not clearly written and the manuscript is not easy to read and understand. I recommend a full editing of the text.
- Page 2 , lane 34: TG content was determined by oil red o quantification. But on legend to Fig 1B,C, it is stated that TG were determined by Sudan II staining. Authors should please correct.
- The name of the protein TACE is only defined in section 2.3. It Is referred previously on the title and abstract and should be defined there.
Author Response
Response to Reviewer’s comment 2
In the present study, Park PJ and colleagues analyzed the role of Cavin-3, a caveolae-associated protein, on lipogenesis and adipogenesis. Furthermore, Cavin-3 modulation of TACE-mediated Pref-1 shedding was explored by the authors. For this, authors developed 2 cell lines of 3T3-L1 adipocytes, one with silencing of Cavin-3 expression and the other overexpressing the protein. Overall, this is an important and interesting subject, but the study is not carefully presented, and some conclusions taken by the authors are not fully supported by the experimental data.
Comments:
In Introduction section essential information about Pref-1 and TACE is lacking.
Q1. Page 2, lane 29: authors state that Cavin-3 levels reach a maximum at day 5 of differentiation and levels are sustained until day 8. However, Fig1A and the western-blotting quantification presented by the authors show more cavin-3 at day 8 comparing to day 5.
A1. Thank you for your comment. In a given experiment, there were no statistical differences of Cavin-3 levels between at 5 and at 8 day of differentiation (Figure S4A). However, the western blot data showed a gradual increase until 8 days. In light of this, we revised the sentence as follows;
|
[Original version] Cavin-3 levels gradually increased, reaching a maximum at day 5 and this level was sustained until day 8. |
[Revised version] - Lane 40-41 in page 2 Cavin-3 levels gradually increased, and continued to do so until day 8. |
Q2. The characterization of the siCavin-3 and over-Cavin-3 cell lines is described on page 3, lanes 13-28 and Fig2A. This should be presented behind data on fig 1B-D, that already show results with those cell lines. Additionally, no results of Fig 2A are described on the text, however RT-qPCR included in supplementary data are fully described. I suggest including Supp Fig S1 A and B on the manuscript.
A2. Thank you for your comment. We also agree with your opinion. So, we explained data and revised our manuscript as follows;
[Revised version]
1) Lane 9-13 in page 3,
Protein expression of Cavin-3 clearly increased on day 8, compared to day 0 of differentiation, in the control groups (normal 3T3-L1, Mock and siControl). In over-Cavin-3 cells, Cavin-3 protein expression was significantly greater than in Mock cells on day 0, which increased by day 8. Cavin-3 protein expression was not detected in siCavin-3 cells at any time-point (Figure 2A).
2) Lane 17-18 in page 3,
Additionally, the expression of FAS and aP2 in the late stage of adipogenesis was also gradually suppressed in siCavin-3 cells (Figure S1A, center and right panel).
3) Lane 21-24 in page 3,
Interestingly, FAS and SCD-1 in over-Cavin-3 cells showed a rapid increase in gene expression during the differentiation period, unlike in the Mock cells (Figure S1B, center and right panel).
Q3. Fig2A: normalization with housekeeping gene is lacking
A3. As requested, we added housekeeping gene and displayed blots in a single gel again (Figure 2A).
[Original version] [Revised version]
Q4. Regarding the cell line overexpressing cavin-3, RT-qPCR data analysis of Fig Supp S1B show no differences on mRNA expression at day 5 and 8. Opposite results were obtained by WB. Authors should comment, please.
A4. Thank you for your comment. We also agree with reviewer’s comment. After receiving questions, we’ve discussed and considered why Mock and over-Cavin-3 cells showed difference in mRNA levels and expression levels of Cavin-3. Based on repeated experiments, we confirmed that the gap between expression levels in Mock and over-Cavin-3 cells was gradually decreased during the differentiation and there were almost no significant differences on day 8. Comparing to the control, the expression level was clearly different on day 8. Based on these repeated results, we speculated that the expression level might reach the saturation level as time goes by since the expression was strongly promoted at the beginning of differentiation. Unlike mRNA levels, the protein level of cavin-3 was rarely maintained as much as the difference as shown in Figure 2A, presumably due to the accumulated amount of expression for 8 days.
Q5. The statistical analysis of all data should be clarified. It is not defined if the comparisons are performed between the day 0 and the different time points of the same condition or between different conditions at the same time point.
A5. Thank you for your comment. As requested, we added the quantitative data in Supporting Information (Figure S4) and highlighted this information at the end of session 4.8 as follows;
[Revised version]
In session 4.8,
Quantitative analysis of all protein levels are presented in Figure S4.
[Additional Figure S4]
Q6. Pref-1 is synthesized as a transmembrane protein but processed to generate soluble forms, including a large 50-kDa soluble form and additional small soluble forms. Several papers claim that only the large soluble form, but not the small soluble or the transmembrane forms of Pref-1, is biologically active to inhibit adipogenesis. Do authors have evidence of biological role of the 21kDa form of Pref-1 detected on the manuscript? Did the authors also detect the 50kDa form?
A6. Thank you for your comment. According to reviewer’s opinion, it was well-known that Pref-1 consists of a large fragment form (50 kDa) and a small fragment form (21 kDa). Prior to our experiments, we purchased the DLK1 antibody from Cell Signaling Technology (Cat. # 2069S, MA, USA), which can detect both a small and a large fragment according to the guideline. We use the DLK1 antibody in our experiments. We were interested in a small fragment rather than a large fragment; the latter was not experimentally confirmed in this study.
[Additional Information]
https://www.cellsignal.com/products/primary-antibodies/dlk1-antibody/2069
Q7. Timp3 action should be validated by analysis of Pref-1 cleavage.
A7. Thank you for your suggestion. We confirmed whether a small fragment was generated by Pref-1 cleavage after treatment of Timp3; we added the result in Supporting Information (Figure S2E).
In the manuscript, we explained that lipolysis was stimulated by the treatment of timp3, a blocking agent of TACE activity; a small fragment of Pref-1was not detected in all cells including siCavin-3 cells when timp3 was treated. Based on this result, we believe that all cells have undergone lipolysis because of normal role of TACE. In addition, each level of adipogenesis was examined by Oil Red O staining and TG analysis as shown in Figure 3B and 3C.
Q8. Page 6, lane 18: “Cavin-3 is located at the inner membrane of mature adipocytes”. What inner membrane are the authors referring to? Furthermore, the immunofluorescence images of cavin-3 (Fig4) suggest a location surrounding the lipid droplets or in the cytosol of the adipocyte. The location at the cell membrane is not clear. The authors could perform a co-localization study using a classic cell membrane marker.
A8. Thank you for your comment. We also agree with your opinion. As requested, we attempted to prepare samples and perform that Cavin-3 is located at the inner membrane of mature adipocytes. Unfortunately, we could not use confocal microscope since it was not available to use due to the COVID-19 issue; we could access and use in other institutions which was not allowed in this situation Therefore, we could not prove the existence of Cavin-3 at the inner membrane. Instead, we revised the manuscript as follows;
|
[Original version] Based on these results, it was confirmed that Cavin-3 is sufficiently expressed and is especially located at the innermembrane of mature adipocytes at day 8 (Figure 4A). |
[Revised version] - Lane 4-5 in page 7 The results showed that Cavin-3 is sufficiently expressed in differentiated adipocytes and is located close to the cytosol membrane (Figure 4A). |
Q9. By WB, Pref-1 is almost undetectable at day 8 of differentiation. In fact, it is known that Pref1 expression inhibition is crucial for the adipocyte differentiation. However, in Fig4 Pref-1 immunofluorescence detection levels are similar at day 0 and day 8. The authors should comment, please. In addition, Pref-1 location does not seem to be the plasma membrane. Again, a co-localization study with a plasma membrane marker could help.
A9. Thank you for your comment. We also examined the Pref-1 level including mRNA and protein expression. As a result, we confirmed that Pref-1 gene expression was reduced in normal 3T3-L1 and also checked that the protein level of small fragment of Pref-1 was larely detected in Figure 2B, 2C and 3A. However, immunofluorescence detection data demonstarted that Pref-1 expression was maintained after adipogenesis; it was distributed around nucleus in immature state, whereas it was localized in cytosol membrane 8 days in mature state as shown in Figure 4B and 4C. As mentioned in the manuscript, we believe that the Pref-1 fragment accumulated during the undifferentiation might not be translocated to cytosol due to increase of Cavin-3, presumably leading to accumulation of full fragment in cell membrane.
As requested, we also agree with reviewer’s opinion; a confocal microscope for immunofluorescence detection approach can provide a more clarified data. So, we attempted to prepare a sample for cell confocal measurement, however we stopped. It was not practically available to use cell IFC due to the COVID-19 issue. Since the confocal is not in Amorepacific laboratory as well as Inje University, we need to use it in different institutions. However, we can’t access and use it all in this situation; we really feel sorry for this situation. Based on our data and previous literatures, we believe that lovastatin leads to the suppression of differentiation through the level as well as composition of caveolae regardless of Cavin-3 increase. Our findings suggest a possibility that Cavin-3 changes are likely to affect fat differentiation through TACE-Pref-1 regulation through the effect of caveolae. We apologize for not providing the conforcal data for a better understanding.
Q10. Authors state that caveolar stability is dependent on cavin-3. This manuscript does not include data supporting this result.
A10. Sorry for the inconvenience. According to previous literatures, Caveolin-1 (Cav1) expression is important for a better understanding of caveolae stability (Ref. Briand et al., 2014, Reference #23, Jansen et al., 2008, Reference #43); the Cav1 level was regulated by caveolae stability. To prove this, the detection of Cav1 expression was performed rather than a direct monitoring of caveolae stability. As a result, we confirmed that the Cavin-1/Cav1 expression was reduced and adipogenesis was also suppressed through lovastatin treatment experiment. In light of this, the change of caveolae stability seemed to be associated with adipogenesis. However, caveolae stability related protein expression was suppressed although Cavin-3 expression did not change, demonstrating that the role of Cavin-3 was a supportive effector rather than a direct effector; it is more reasonable and understandable in a given experiment.
|
[Original version] 2.5. Cavin-3-dependent caveolar stability affects Pref-1 cleavage through TACE activity … To clarify whether Cavin-3-dependent caveolar stability affects Pref-1 cleavage through TACE activity, localization changes were analyzed in lovastatin treated over-Cavin-3 by confocal immune-fluorescence microscopy. These experiments showed that Cavin-3 and Pref-1 do not collect and disperse in the same way as those in the undifferentiated state (Figure 4B, upper panel, 5B and Figure S3D). Taken together, our observations suggest that Cavin-3 modulation through gene manipulation affects caveolar stability, which in turn regulates adipocyte differentiation by influencing TACE-mediated Pref-1 shedding/cleavage (Figure 6). |
[Revised version; Lane 14-23 in page 9] 2.5. Cavin-3 supports caveolar stability to inhibit the translocation of Pref-1 small fragment into cytosol … Next, to clarify whether the change of caveolar stability affected Cavin-3, confocal immunofluorescence microscopy was used to investigate changes in localization in lovastatin-treated over-Cavin-3 cells. The disruption in caveolae stability resulted in Cavin-3 expression being dispersed throughout the cytoplasm. Furthermore, Cavin-3 and Pref-1 did not co-localize (Figure 4B, upper panel, 5B and Figure S3D). These results suggest that Cavin-3 modulation by gene manipulation affects caveolar stability by influencing the support role of Cavin-3, thus affecting the regulation of Pref-1 small fragment translocation into the cytosol (Figure 6). |
Q11. In general, the sentences are not clearly written and the manuscript is not easy to read and understand. I recommend a full editing of the text.
: Sorry for the inconvenience. To clarify, we revised and edited our manuscript, which was done by a company named Editage. We attached the certificate of English editing.
Q12. Page 2 , lane 34: TG content was determined by oil red o quantification. But on legend to Fig 1B,C, it is stated that TG were determined by Sudan II staining. Authors should please correct.
: Thank you for your comment. As requested, we revised the legend of figure as follows;
|
[Original version] The TG content was quantified by staining the cells (n = 3) with Sudan II and measuring the absorbance at 490 nm. |
[Revised version] - Lane 45-46 in page 2 Intracellular total triacylglycerol (TG) content was quantified by measuring the absorbance of ORO stained LDs (490 nm). |
Q13. The name of the protein TACE is only defined in section 2.3. It Is referred previously on the title and abstract and should be defined there.
A13. Thank you for your advice. We inserted and highlighted it in 2nd sesseion in introduction (Lane 8 in page 1 to lane 9 in page 2). Also, we rearranged the references depending on the revision of sentences as follows;
“Preadipocyte factor 1 (Pref-1) is highly expressed in preadipocytes and inhibits adipocyte differentiation by maintaining the preadipocyte state. Pref-1 is a transmembrane protein consisting of six epidermal growth factor (EGF)-like repeats, a juxtamembrane region, a single transmembrane domain and a short cytoplasmic tail [8-11]. Full-length Pref-1 is proteolytically cleaved at its extracellular domain by tumor necrosis factor-alpha-converting enzyme (TACE, ADAM17), generating soluble forms of Pref-1 [12-15]. The soluble forms of Pref-1 produced by TACE are reported to inhibit adipocyte differentiation by suppressing factors involved in the early stage of adipogenesis, namely the CCAAT/enhancer-binding protein beta and delta (C/EBPb and C/EBPd), via induction of Sox9 expression through activation of the MEK/ERK pathway [15]. Pref-1 expression decreases gradually during adipocyte differentiation, and decreased Pref-1 expression alone is sufficient to increase adipocyte differentiation [11].”

Round 2
Reviewer 1 Report
The manuscript is now basically fine, the authors have attempted to address all the concerns raised. However, for understandable practical reasons, they have not been able to use TEM to determine that lovastatin is effective in disrupting caveola formation. Therefore, in the section of the manuscript where they describe the data from Fig 5, they need to qualify their conclusions with the fact that they were unable to verify that the indicated treatment with lovastatin triggered a loss of detectable caveolae.
Author Response
A1. Thank you for your understanding. We also agree with your opinion. To convey more clearly, we revised our manuscript as follows;
|
[Original version] - Lane 6-10 in page 9 The drug treatment results showed that despite the increment of Cavin-3 protein expression depend on the adipogenesis process, the expression of caveolae stability and adipogenesis facilitated signals, such as Cavin-1, Cav-1 and PPARg, decreased with increasing expression of Pref-1, at which time the TACE did not change. Therefore, we assume that caveolar stability could induce changes in adipogenesis-related factors such as Pref-1 and PPARg. |
[Revised version] - Lane 6-14 in page 9 The drug treatment results showed that despite the increment of Cavin-3 protein expression depend on the adipogenesis process, the expression of caveolae stability-related and adipogenesis facilitated signals, such as Cavin-1, Cav-1 and PPARg, decreased with increasing expression of Pref-1, at which time the TACE did not change. Although the results did not directly confirm the disrupting caveolae formation by lovastatin treatment, previous study has shown that changes in the caveolar stability could occur due to the disruption of caveolae formation caused by a drug like lovastatin [43]. Therefore, we assume that changes in the caveolar stability by lovastatin could induce changes in adipogenesis-related factors such as Pref-1 and PPARg. |

Reviewer 2 Report
I appreciate the author’s effort on the manuscript revision however, I consider that was not enough for publication. Some points were not answered, and others not attended due to understandable issues related with the Covid-9 pandemic. But, if not possible to perform some additional experiments, the authors should revise the paper and present only the information fully supported by the experimental data. Furthermore, the study is still not carefully presented and the English needs further revision.
Comments:
1-Previous comment: Page 2, lane 29: authors state that Cavin-3 levels reach a maximum at day 5 of differentiation and levels are sustained until day 8. However, Fig1A and the western-blotting quantification presented by the authors show more cavin-3 at day 8 comparing to day 5.
I do not understand why authors decided to describe in the text that the cavin-3 expression levels increase until day 8 while the quantification of another experiment (Fig S4A) revealed no statistical differences between day 5 and 8. The authors should repeat the WB experiments with different cell samples to have sure of the results. And the WB figure should be representative of the final data, which comprises the quantification of all individual WBs.
2- In section 2.2, line 117 and line 133 start the description of covin-3 effect on adipogenesis. These should be presented together and not be divided. Lines 119-120: figures 2B and 2C do not shown expression levels of FAS or aP2. They are presented only in Fig S1A and S1B. As suggested previously I think that they should be included on the manuscript since this data is intensively described in the text.
3- Previous comment: The statistical analysis of all data should be clarified. It is not defined if the comparisons are performed between the day 0 and the different time points of the same condition or between different conditions at the same time point.
This point was not fully answered. The detailed description of the statistical analysis should be incorporated in the manuscript. For example, in FigS1A and S1b, which experimental groups are compared?
4- Previous comment: Pref-1 is synthesized as a transmembrane protein but processed to generate soluble forms, including a large 50-kDa soluble form and additional small soluble forms. Several papers claim that only the large soluble form, but not the small soluble or the transmembrane forms of Pref-1, is biologically active to inhibit adipogenesis. Do authors have evidence of biological role of the 21kDa form of Pref-1 detected on the manuscript? Did the authors also detect the 50kDa form?
This point was not answered. Authors stated that they are interested in the small soluble fragment and that the antibody recognize both the large and the small fragment, but no explanation or reason was presented for this selection.
5- Previous comment: Page 6, lane 18: “Cavin-3 is located at the inner membrane of mature adipocytes”. What inner membrane are the authors referring to? Furthermore, the immunofluorescence images of cavin-3 (Fig4) suggest a location surrounding the lipid droplets or in the cytosol of the adipocyte. The location at the cell membrane is not clear. The authors could perform a co-localization study using a classic cell membrane marker. In addition, Pref-1 location does not seem to be the plasma membrane. Again, a co-localization study with a plasma membrane marker could help.
I understand the difficulties in performing additional experiments using the confocal microscope, but the text should be revised and results supported by the existing images. The authors state now that Cavin-3 is located close to the cytosol membrane. In cell biology there is no “cytosol membrane”! Do authors refer to cell membrane, or the inner/cytosolic side of the cell membrane? However, the images also do not demonstrate this…and in the absence of co-localization studies the conclusions should be carefully taken. The same for Pref-1 location.
6- Previous comment: Page 2 , lane 34: TG content was determined by oil red o quantification. But on legend to Fig 1B,C, it is stated that TG were determined by Sudan II staining.
The author revised this information on the figure legend and state that ORO was the used method but on methods section 4.7 TG content is measured by Sudan II method. Additionally, ORO staining measures all the lipid content of the adipocyte and not only TG, thus is not an appropriate methodology for assessment of TG levels.
Author Response
Reviewer 2
; Thank you for your comment. Also, the authors appreciates for your understanding, so revised manuscript which was based on the experimental data; the author focused on facts than expected.
Comments:
Q1-Previous comment: Page 2, lane 29: authors state that Cavin-3 levels reach a maximum at day 5 of differentiation and levels are sustained until day 8. However, Fig1A and the western-blotting quantification presented by the authors show more cavin-3 at day 8 comparing to day 5.
I do not understand why authors decided to describe in the text that the cavin-3 expression levels increase until day 8 while the quantification of another experiment (Fig S4A) revealed no statistical differences between day 5 and 8. The authors should repeat the WB experiments with different cell samples to have sure of the results. And the WB figure should be representative of the final data, which comprises the quantification of all individual WBs.
A1. Thank you for your comment. The author additionally performed the western blotting for comparing the protein expression levels. As a result, we confirmed that there is a statistical difference between on day 5 and on day 8. We added these figures in Figure 1A and Figure S4A.
Q2- In section 2.2, line 117 and line 133 start the description of Cavin-3 effect on adipogenesis. These should be presented together and not be divided. Lines 119-120: figures 2B and 2C do not shown expression levels of FAS or aP2. They are presented only in Fig S1A and S1B. As suggested previously I think that they should be included on the manuscript since this data is intensively described in the text.
A2. Thank you for your comment. Based on your opinion, the author revised and rearranged the figures; inserted Fig. S1A and Fig. S1B into Figure 2 for a better understanding as follows.
Additionally, the related manuscript has been revised as follows.
|
[Original version] - Section 2.2 in page 3 The expression levels of Cavin-3 and adipogenesis-related genes, including fatty acid synthase (FAS), adipocyte protein 2 (aP2) and stearoyl-CoA desaturase (SCD-1) were determined by real-time quantitative polymerase chain reaction (RT-qPCR) during adipogenesis periods (Figure 2B, 2C and Figure S1A and S1B). In siCavin-3 cells, Cavin-3 mRNA showed the initial inhibition was less than 30%, but markedly decreased approximately 70% after 8 days of differentiation. Additionally, the expression of FAS and aP2 in the late stage of adipogenesis was also gradually suppressed in siCavin-3 cells (Figure S1A, center and right panel). Cavin-3 expression levels were significantly different, approximately 2.2 times, between over-Cavin-3 and mock-transfected cells at the beginning of differentiation (day 0). However, the expression gap rates were gradually reduced and showed no difference by day 8 (Figure S1B, left panel). Interestingly, FAS and SCD-1 in over-Cavin-3 cells showed a rapid increase in gene expression during the differentiation period, unlike in the Mock cells (Figure S1B, center and right panel). There was no statistical difference in ORO stained LDs and TG content (Figure S1C) between control cells, which included siControl, Mock, and non-transfected 3T3-L1 cells. Taken together Based on these results, it was confirmed that the stable cell lines promoting/suppressing Cavin-3 gene expression significantly regulated the increase/decrease of Cavin-3 gene expression, respectively. Next, to determine whether Cavin-3 affects adipogenesis-related gene expression, we verified the mRNA expression levels of preadipocyte factor 1 (Pref-1), peroxisome proliferator-activated receptor gamma (PPAR and adiponectin (Adipoq) by RT-qPCR at various time points after differentiation. Pref-1 mRNA expression was not markedly increased during adipocyte differentiation in siControl cells, but sharply increased in siCavin-3 cells, even after the induction of differentiation (Figure 2B, left panel). In contrast, the mRNA expression levels of PPAR and Adipoq, two well-known adipocyte differentiation markers [33, 34], were suppressed in siCavin-3 cells, however, they were not changed in siControl cells (Figure 2B, center and right panels). In contrast to siCavin-3 cells, the expression level of Pref-1 in over-Cavin-3 cells was similar to mock transfected cells (Figure 2C, left panel). While PPAR and Adipoq gene expression showed similar patterns in over-Cavin-3 and mock-transfected cells at day 2, they significantly increased in over-Cavin-3 cells compared to mock cells thereafter (Figure 2C, center and right panels). We also monitored the expression levels of FAS, aP2, SCD-1, and CCAAT/enhancer-binding protein beta and delta (C/EBPβ and C/EBPδ) (Figure S1A, S1B and S1D) [35-37]. Based on these additional results, it was confirmed that genes in later phases of adipocyte differentiation were also affected by Cavin-3 gene modulation. Overall, these results suggest that changes in Cavin-3 gene expression are linked to the regulation of adipogenesis-related gene expression. |
[Revised version] - Section 2.2 in page 9 The expression levels of Cavin-3 and adipogenesis-related genes, including preadipocyte factor 1 (Pref-1), peroxisome proliferator-activated receptor gamma (PPARg) and adiponectin (Adipoq), fatty acid synthase (FAS), adipocyte protein 2 (aP2) and stearoyl-CoA desaturase (SCD-1) were determined by real-time quantitative polymerase chain reaction (RT-qPCR) during adipogenesis, respectively (Figure 2B-E). In siCavin-3 cells, Cavin-3 mRNA showed the initial inhibition was less than 30%, but markedly decreased approximately 70% after 8 days of differentiation (Figure 2B, left panel). Additionally, to determine whether Cavin-3 affects adipogenesis-related gene expression, we verified the mRNA expression levels of Pref-1, PPARg, Adipoq, FAS and aP2 at various time points after differentiation. As results, Pref-1 mRNA expression was gradually decreased during adipocyte differentiation in siControl cells, but sharply increased in siCavin-3 cells, even after the induction of differentiation (Figure 2B, center panel). In contrast, the mRNA expression levels of PPARg and Adipoq, two well-known adipocyte differentiation markers [33, 34], FAS and aP2 in the late stage of adipogenesis, were suppressed in siCavin-3 cells (Figure 2B, right panel and 2C). In over-Cavin-3 cells, Cavin-3 expression levels were significantly different, approximately 2.2 times, between over-Cavin-3 and mock-transfected cells at the beginning of differentiation (day 0) (Figure 2D, left panel). However, the expression gap rates were gradually reduced and showed no difference by day 8. In contrast to siCavin-3 cells, the expression level of Pref-1 in over-Cavin-3 cells was similar to mock transfected cells (Figure 2D, center panel). Interestingly, PPARg, Adipoq, FAS and SCD-1 in over-Cavin-3 cells showed a rapid increase in gene expression during the differentiation period, unlike in the Mock cells (Figure 2D, right panel and 2E). While PPARg, Adipoq, FAS and SCD-1 gene expression showed similar patterns in over-Cavin-3 and mock-transfected cells on day 2, they significantly increased in over-Cavin-3 cells compared to mock cells thereafter. There was no statistical difference in ORO stained LDs and TG content (Figure S1A and S1B) among the control cells, which included siControl, Mock, and non-transfected 3T3-L1 cells. Taken together Based on these results, it was confirmed that the stable cell lines promoting/suppressing Cavin-3 gene expression significantly regulated the increase/decrease of Cavin-3 gene expression, respectively. We also monitored the expression levels of CCAAT/enhancer-binding protein beta and delta (C/EBPβ and C/EBPδ) (Figure S1C and S1D) [35-37]. Based on these additional results, it was confirmed that genes in later phases of adipocyte differentiation were also affected by Cavin-3 gene modulation. Overall, these results suggest that changes in Cavin-3 gene expression are linked to the regulation of adipogenesis-related gene expression. |
Q3- Previous comment: The statistical analysis of all data should be clarified. It is not defined if the comparisons are performed between the day 0 and the different time points of the same condition or between different conditions at the same time point.
This point was not fully answered. The detailed description of the statistical analysis should be incorporated in the manuscript. For example, in FigS1A and S1b, which experimental groups are compared?
A3. Thank you for highlighting this important matter. In order to clarify the statistical analysis, the text from the figure legend 2 was thoughtfully changed. Additionally, after receiving the reviewer’s question, the authors came to a conclusion that not only the analysis should be more clearly explained but also it should be complemented. Therefore, two analyses were performed; the original one where the statistical relevance of the gene expression changes, through the different days (2, 5 and 8), were compared to the day 0, marked with an asterisk (*). And, the statistical difference between the different groups at the same time points (siControl versus siCavin-3 and Mock versus over-Cavin-3) were marked with a star cross (†). The legend of Figure 2 changed as follows;
|
[Original version] - Figure legend 2 Figure 2. Cavin-3 gene modulation influences adipogenesis-related mRNA expressions. (A) Expression of Cavin-3 protein was validated in 3T3-L1 adipocytes, siControl, siCavin-3, Mock and over-Cavin-3 cells by western blot analysis at day 0 and 8 of differentiation. g-TUB was used as a control and the grouping of blots cropped from different gels. (B, C) The mRNA expression of Pref-1, PPARg, and Adipoq in siControl (n = 3) vs siCavin-3 (n = 3) and Mock (n = 3) vs over-Cavin-3 (n = 3) cells, respectively. The expression levels were determined by RT-qPCR at various time points (0, 2, 5 and 8 days of differentiation). The data are presented as the means ± SD. (*P< 0.05, **P< 0.01, ***P< 0.001; data were analyzed with an Unpaired Student’s t-test). |
[Revised version] - Figure legend 2 Figure 2. Cavin-3 gene modulation influences adipogenesis-related mRNA expressions. (A) Expression of Cavin-3 protein was validated in 3T3-L1 adipocytes, siControl, siCavin-3, Mock and over-Cavin-3 cells by western blot analysis at day 0 and 8 of differentiation. g-TUB was used as a control and the grouping of blots cropped from different gels. (B-E) The mRNA expression of Cavin-3, Pref-1, PPARg, Adipoq, FAS, aP2 and SCD-1 in siControl (n = 3) vs siCavin-3 (n = 3) and Mock (n = 3) vs over-Cavin-3 (n = 3) cells, respectively. The expression levels were determined by RT-qPCR at various time points (0, 2, 5 and 8 days of differentiation). The data are presented as the means ± SD. (*P< 0.05, **P< 0.01, ***P< 0.001 compared to siContol or Mock at day 0: †P< 0.05, ††P< 0.01, †††P< 0.001 compared to siControl or Mock at each time point, respectively; data were analyzed with an Unpaired Student’s t-test). |
On the “Section 2.2 in Result”, it was highlighted which statistics was used to make each conclusion.
|
[Revised version] - Lane 4 in page 4 In over-Cavin-3 cells, Cavin-3 expression levels were significantly different, approximately 2.2 times (††P< 0.01), between over-Cavin-3 and mock-transfected cells at the beginning of differentiation (day 0) (Figure 2D, left panel). |
[Revised version] - Lane 10 in page 4 While PPARg, Adipoq, FAS and SCD-1 gene expression showed similar patterns in over-Cavin-3 and mock-transfected cells on day 2, they significantly increased (†P< 0.05, ††P< 0.01, †††P< 0.001) in over-Cavin-3 cells compared to mock cells thereafter. |
Q4- Previous comment: Pref-1 is synthesized as a transmembrane protein but processed to generate soluble forms, including a large 50-kDa soluble form and additional small soluble forms. Several papers claim that only the large soluble form, but not the small soluble or the transmembrane forms of Pref-1, is biologically active to inhibit adipogenesis. Do authors have evidence of biological role of the 21kDa form of Pref-1 detected on the manuscript? Did the authors also detect the 50kDa form?
This point was not answered. Authors stated that they are interested in the small soluble fragment and that the antibody recognize both the large and the small fragment, but no explanation or reason was presented for this selection.
A4. Thank you for your comment. According to previous literatures, it was well-known that Pref-1 is an important factor in the early stage of adipogenesis and that full length Pref-1 was existed in cell membrane. Briefly, the cellular mechanism is as follows. Extracellular domain of Pref-1 is cleaved by TACE, then small fragment of Pref-1 is moved into nucleus and also inhibits the expressions of C/EBPb and C/EBPd, leading to inhibition of adipogenesis process. In addition, it was also well-known that the expression level of Pref-1 was reduced during adipogenesis (Smas et al., 1997, Lee et al., 2003, Wang et al., 2006, Sul HS, 2009, Unger RH, 2009 [11-15]). Based on these literatures, the authors revised and highlighted Pref-1 related parts as follows;
[ 2nd paragraph in Introduction]
Preadipocyte factor 1 (Pref-1) is highly expressed in preadipocytes and inhibits adipocyte differentiation by maintaining the preadipocyte state. Pref-1 is a transmembrane protein consisting of six epidermal growth factor (EGF)-like repeats, a juxtamembrane region, a single transmembrane domain and a short cytoplasmic tail [8-11]. Full-length Pref-1 is proteolytically cleaved at its extracellular domain by tumor necrosis factor-alpha-converting enzyme (TACE, ADAM17), generating soluble forms of Pref-1 [12-15]. The soluble forms of Pref-1 produced by TACE are reported to inhibit adipocyte differentiation by suppressing factors involved in the early stage of adipogenesis, namely the CCAAT/enhancer-binding protein beta and delta (C/EBPb and C/EBPd), via induction of Sox9 expression through activation of the MEK/ERK pathway [15]. Pref-1 expression decreases gradually during adipocyte differentiation, and decreased Pref-1 expression alone is sufficient to increase adipocyte differentiation [11].]
The authors believed that changes of Pref-1 expression regulated by Cavin-3 plays an important role in adipogenesis as a trigger. Based on previous literatures combined with our results, the authors believe that Cavin-3 affect the small fragment of Pref-1, leading to regulation of adipogenesis. Therefore, from the beginning, we focused on such genes and protein expression, furthermore their locations in adipocytes. In light of this, we revised and added some sentences as follows;
|
[Original version] -2.3 section, lane 7~ in Page 5 To determine whether Cavin-3 is involved in controlling the TACE-mediated Pref-1 cleavage process, Pref-1 shedding results were validated using western blot analysis in normal 3T3-L1 adipocyte and Cavin-3 modulated stable cell lines (Figure 3A). Pref-1 was detected on day 0 in 3T3-L1 adipocytes, but, the smaller band (21 kDa) was weakly present in 3T3-L1 adipocytes, siControl, Mock and over-Cavin-3 at day 8. |
[Revised version] - 2.3 section, lane 7~13in Page 6 To determine whether Cavin-3 is involved in controlling the TACE-mediated Pref-1 cleavage process, Pref-1 shedding results were validated using western blot analysis in normal 3T3-L1 adipocyte and Cavin-3 modulated stable cell lines (Figure 3A). In particular, we focused on the expression level change of Pref-1 small fragment, which has known to be involved in the adipogenesis. As a result, the smaller band of Pref-1 (21 kDa) was detected on day 0 in 3T3-L1 adipocytes, but, weakly present in 3T3-L1 adipocytes, siControl, Mock and over-Cavin-3 at day 8. |
Q5- Previous comment: Page 6, lane 18: “Cavin-3 is located at the inner membrane of mature adipocytes”. What inner membrane are the authors referring to? Furthermore, the immunofluorescence images of cavin-3 (Fig4) suggest a location surrounding the lipid droplets or in the cytosol of the adipocyte. The location at the cell membrane is not clear. The authors could perform a co-localization study using a classic cell membrane marker. In addition, Pref-1 location does not seem to be the plasma membrane. Again, a co-localization study with a plasma membrane marker could help.
I understand the difficulties in performing additional experiments using the confocal microscope, but the text should be revised and results supported by the existing images. The authors state now that Cavin-3 is located close to the cytosol membrane. In cell biology there is no “cytosol membrane”! Do authors refer to cell membrane, or the inner/cytosolic side of the cell membrane? However, the images also do not demonstrate this…and in the absence of co-localization studies the conclusions should be carefully taken. The same for Pref-1 location.
A5. Thank you for your comment. The author also agrees with your opinion. The manuscript should contain what the data can support, helping broad readers understand while reducing the ambiguous explanation. In light of this, we revised our manuscript as follows;
|
[Original version] - 2.4 section in Result To validate how Cavin-3 affects TACE-mediated Pref-1 shedding, the localization and physiological changes of Cavin-3 and Pref-1 were observed using confocal immunofluorescence microscopy (Figure 4). The results showed that Cavin-3 is sufficiently expressed in differentiated adipocytes and is located close to the cytosol membrane (Figure 4A). There was a noticeable difference in expression and localization of Pref-1 and Cavin-3 pre- and post-maturation (Figure 4B). On day 0, Pref-1 seemed to spread throughout the cell, except in the nucleus, and Cavin-3 was rarely expressed (Figure 4B, upper panel). On day 8, Pref-1 was almost laid in cell membrane, even though the cell size was increased after differentiation (Figure 4B, lower panel). Notably, as Cavin-3 localization increased and concentrated around the cell membrane, Pref-1 expression also concentrated around the cell membrane. |
[Revised version] - 2.4 section in Result To validate how Cavin-3 affects TACE-mediated Pref-1 shedding, the localization and physiological changes of Cavin-3 and Pref-1 were observed using confocal immunofluorescence microscopy (Figure 4). As a result, Cavin-3 appeared to be sufficiently expressed in the differentiated adipocytes and located as if covering the stained lipid droplets in the cells. (Figure 4A). There was a noticeable difference in expression and localization of Pref-1 and Cavin-3 pre- and post-maturation (Figure 4B). On day 0, Pref-1 seemed to spread throughout the cell, except in the nucleus, and Cavin-3 was rarely expressed (Figure 4B, upper panel). On day 8, although no further positioning was confirmed using plasma membrane markers, Pref-1 seemed to be continuously detected and appeared to be distributed outside the cell, as shown in the previous study about the location of full frame sized Pref-1- [12-15], even though the cell size was increased after differentiation (Figure 4B, lower panel). Notably, as Cavin-3 localization increased and concentrated around the lipid droplets, Pref-1 expression concentrated around the cell membrane. |
Q6- Previous comment: Page 2, lane 34: TG content was determined by oil red o quantification. But on legend to Fig 1B, C, it is stated that TG were determined by Sudan II staining.
The author revised this information on the figure legend and state that ORO was the used method but on methods section 4.7 TG content is measured by Sudan II method. Additionally, ORO staining measures all the lipid content of the adipocyte and not only TG, thus is not an appropriate methodology for assessment of TG levels.
A6. Thank you for your comment. In our experiment, the author performed the staining assay (n=3) as well as TG content assay (n=3) under the same condition. Staining was visualized by Oil red O staining dye, whereas the quantitative analysis of TG content was performed by Sundan Ⅱ under the same condition. To clarify this, we revised and explained more in Materials and Methods 4.6 and 4.7 as follows;
[4.6. Oil Red O staining
Oil Red O (ORO; Sigma-Aldrich) stock solution was prepared by dissolving ORO in 60% propylene glycol (PG) (Santa Cruz Biotechnology, CA, USA), and used to highlight the amount of lipid droplet in differentiated 3T3-L1 adipocytes. On 0, 2, 5, and 8 days after differentiation induction, 3T3-L1 cells were washed twice with cold phosphate-buffered saline (PBS; Welgene, Daegu, Korea), before being fixed with 3.7% formaldehyde (Sigma-Aldrich) in PBS for 1 h. The fixed cells were washed with PG and stained with ORO working solution (0.3% Oil Red O in 60% PG) for 30 min. The cells were then washed three times with 85% PG and rinsed gently with tap water. The phenotypic changes in the cells were visualized and photographed using an Olympus IX71 microscope (Tokyo, Japan).
4.7. Analysis of TG content
The amount of accumulated TG was measured using the Sudan II (Sigma-Aldrich) staining method. The cells were washed twice in cold PBS, before being fixed with 3.7% formaldehyde for 20 min. Cells were then stained with 0.5% Sudan II in 60% isopropyl alcohol (Sigma-Aldrich) for 1 h. The stained cells were washed gently with 70% ethyl alcohol (Sigma-Aldrich), and the TG-bound Sudan II was extracted with 4% NP-40 (Sigma-Aldrich) in isopropyl alcohol for 20 min. A Synergy H2 spectrophotometer (BioTek, VT, USA) was used to measure the extract absorbance at 490 nm. ]
Also, the author revised some sentences for a better understanding as follows;
[Revised version, 2.1 section in page 2]
On day 8, lipid droplets (LDs), a physiological maturation marker in adipocytes, were visualized by Oil Red O (ORO) staining. Intracellular total triacylglycerol (TG) content was quantified by measuring the absorbance using Sudan II (490 nm).
[Revised version, legend part of Figure 2]
(B, C) The Oil Red O (ORO) staining and total glyceride (TG) content were validated in siCavin-3 and over-Cavin-3 cells after 8 and 5 days of differentiation, respectively. Scale bar = 200 μm. The TG content was quantified by Sudan II staining the cells (n = 3) by measuring the absorbance at 490 nm.
